# Detection Method of Fry Feeding Status Based on YOLO Lightweight Network by Shallow Underwater Images

**Haihui Yang** , **Yinyan Shi and Xiaochan Wang** *

College of Engineering, Nanjing Agricultural University, Nanjing 210031, China
* Correspondence: wangxiaochan@njau.edu.cn

**Abstract:** Pellet feed is widely used in fry feeding, which cannot sink to the bottom in a short time, so most fries eat in shallow underwater areas. Aiming at the characteristics of fry feeding, we present herein a nondestructive and rapid detection method based on a shallow underwater imaging system and deep learning framework to obtain fry feeding status. Towards this end, images of fry feeding in shallow underwater areas and floating uneaten pellets were captured, following which they were processed to reduce noise and enhance data information. Two characteristics were defined to reflect fry feeding behavior, and a YOLOv4-Tiny-ECA network was used to detect them. The experimental results indicate that the network works well, with a detection speed of 108FPS and a model size of 22.7 MB. Compared with other outstanding detection networks, the YOLOv4-Tiny-ECA network is better, faster, and has stronger robustness in conditions of sunny, cloudy, and bubbles. It indicates that the proposed method can provide technical support for intelligent feeding in factory fry breeding with natural light.

**Keywords:** aquaculture; YOLO lightweight network; fry feeding status detection; shallow underwater images

## 1. Introduction

Since 2020, global fishing production has exceeded 178 million tons; humans consumed 88% of this output [1]. As the world's population continues to expand, the strain on the global fisheries market will also grow [2]. From January 2020, China began to implement 'The Fishing Ban in Yangtze River' for ten years, and productive exploitation of natural fishery resources shall be prohibited during that period [3–5]. Increasing breeding species, reforming breeding technologies, and promoting advanced breeding patterns are becoming the priority to develop, and factory aquaculture has become the current popular model [6,7].

Fry breeding is an important part of factory aquaculture, and high-quality fry is the foundation to support the development of fisheries; healthy fry directly affects the yield and economic benefits [8]. At present, most of the fry factories in China adopt the feeding mode of manual feeding or timed and quantitative machine feeding. In manual feeding, workers determine the feeding amount according to experience and eye observation, which is characterized by high labor intensity and low accuracy, and most of the feeding machines in the market are not intelligent enough to determine the feeding quantity of fry [9,10]. The current feeding method easily causes underfeeding or overfeeding; underfeeding will lead to fish nutrition deficiency, and overfeeding will cause the waste of resources and pollute the water. The feed cost accounts for more than 60% of the variable cost in fry breeding [11]. Hence the accurate and effective detection of fry feeding status is very necessary.

In previous studies, researchers used a variety of sensors to obtain fish status. With the improvement of computer vision technology, its convenience and non-damage characteristics make it widely used in fish behavior recognition [12].

Chen et al. [13] combined the texture method and area method to study the fish feeding behavior; four texture features of the inverse moment, correlation, energy, and contrast

were analyzed in the study. The results showed that the contrast and intensity of fish population activity reached 89.42%, while it did not consider the interference caused by splashes and individual overlaps. Guo et al. [14] proposed a method to identify the feeding behavior of shoal carp by calculating the shape parameters and image entropy of the feeding images, and a BP neural network model was used to identify the feeding status of fish. Water surface jitter and water spray were regarded as unique texture characteristics, and the recognition accuracy reached 98.0%. Zhao et al. [15] proposed a nondestructive and effective method based on computer vision that can quantify the feeding intensity of tilapia. This method can quantify the overall feeding intensity of fish with the aid of the Lucas–Kanade optical flow method and information entropy.

Uneaten pellets recognition is another method of using computer vision to analyze, identify, and evaluate the feeding intensity. Cameras are used to acquire images of uneaten pellets after feeding; computers are used to process the image, identify the residual bait, and count and analyze it [16]. When the number of uneaten pellets reaches a certain threshold, it suggests that the overall fish-feeding desire is decreasing, and the controller should reduce or stop the feed supply. Zhou et al. [17] proposed an improved YOLOv4 network, which can detect uneaten pellets of small size and improve the detection performance of the original network by 27.21%. Atoum et al. [18] adopted a two-stage method to automatically detect the amount of excessive feed on the water surface; the results showed that the two-stage method has higher accuracy than the assessment of fish behavior alone.

In view of the above, computer vision has been widely used to analyze fish-feeding behavior, and the improvement of computer hardware performance and the continuous optimization of image processing algorithms have increased the accuracy of fish-feeding behavior recognition [19]. However, most of the methods are still used in the laboratory, and the complexity of the breeding environment and the particularity of the research objects (uncontrollable, individual overlap, fast movement, etc.) make it difficult to acquire high-quality images [20,21]. Intelligent feeding based on fish feeding behavior detection still faces major challenges. Further, researchers usually choose mature fish as the object, and images of feeding behavior and uneaten pellets were captured underwater or overwater. In the practical breeding environment, with different light conditions, the image quality is difficult to control, and the uneaten pellet is difficult to detect [22]. Aiming at the characteristics of fry feeding in shallow underwater, this study defines two typical feeding characteristics: angle feeding and vertical feeding. The fry feeding status is estimated by detecting the two typical feeding characteristics.

Deep learning approaches have proven to be a key technology for intelligent aquaculture systems. Existing convolutional neural networks can be divided into two categories: (1) two-stage methods and (2) one-stage methods. Although the two-stage method has higher accuracy, it is too slow due to intensive computation. The one-stage method has a simpler structure and has been proven to be applicable to the real-time identification of animal behavior [23]. At present, the most widely used one-stage network is the YOLO family network [24–26], which combines the RPN network with target recognition to simplify detection and improve speed; its lightweight characteristics make it possible to deploy on edge devices. Sung et al. [27] proposed an online fish detection method based on YOLO; the detection accuracy is 93%, the intersection over union (IoU) is 0.634, and the detection speed is 16.7FPS. Cai et al. [28] used a YOLOv3 network with the bottleneck of MobileNetv1 to monitor fish in a breeding farm.

The overall goal of this study is to develop an effective detection system for fry feeding behavior, to provide support for intelligent fry breeding, to ensure the welfare of fry feeding, and improve productivity. The feeding status detection in this study is based on the fry feeding characteristics, and it is aimed at the deployment of edge devices, so the lightweight of the detection model is crucial. In this study, the YOLOv4-Tiny network with the attention mechanism of the ECA module is adopted to realize the effective detection of fry feeding status. The underwater imaging system adopted in this study has a simple structure and is easy to operate. Data enhancement and noise reduction are used to preprocess the acquired

images to improve image quality, and the image augmentation method is used to enrich the size of the data set.

The remainder of the paper is organized as follows: Section 2 details the materials, underwater imaging system, and image processing scheme. Section 3 gives the results and discusses them. Finally, Section 4 summarizes and prospects the whole paper.

## 2. Materials and Methods

### 2.1. Material and Sample Preparation

The experiment was performed at the Institute of Technology, Nanjing Agricultural University (Nanjing, China). Altogether, 200 grass carp fry were used in this study, with a mean weight of 5.5 g and a mean body length of 7.4 cm. The fries were selected because of their strong adaptability and their wide use in aquaculture in China. The fries were purchased from Zhenyuan Agricultural Development Co., Ltd. (Guiyang, China) and were delivered by express delivery within 72 h after being caught out of water. Nowadays, graded breeding is widely used in factory fry farming; that is, fries are gradually evacuated and separately raised based on body length growth. After about 20 days of breeding, the fry body length can reach 7 cm, and they can start to develop feeding habits—the research object of this study is grass carp fry at this stage.

After the arrival, the fry population was raised in the fishpond for two weeks to adapt to the experiment environment. The pond was made of canvas, with a diameter of 1.5 m and a water depth of 90 cm. The fishpond was sterilized before the fry population was released. In order to protect the breeding environment, the water was exchanged every two days after the fry were released, and the water was exchanged by 50% each time to keep the ammonia nitrogen and nitrite contents in the fishpond at the appropriate value (<0.2 mg/L). Meanwhile, the pH level of the water was kept at $7.5 \pm 0.5$, and the water temperature was kept at $22 \pm 2\ °C$. During the period of fry adaption to the environment, the feeding rate was 2% of body weight per day with a commercial diet. When feeding, the feed required for a single time will be evenly distributed on the water surface, and the uneaten pellets will be pulled out after 2 h to prevent water pollution.

### 2.2. Experimental System and Data Acquisition

The experimental system is installed in a glass greenhouse on the roof of the Boyuan Building of Nanjing Agricultural University. As shown in Figure 1, the experimental system consisted mainly of a fishpond, a filtration system, a temperature control system, and a feeding system. The filtering system and temperature control system were connected to the fishpond through the water pipe; the filtering system was used to filter out the harmful material in the fish pond, the temperature control system regulates the water temperature and keeps the water temperature stable, and the feeding system sprinkles feed from above the fish pond to the water surface. The image acquisition system includes 3 SARGO A8 waterproof cameras (SARGO, Inc., Guangzhou, China) with 20 million pixels, CMOS sensor, and a maximum resolution of 3840 pixels × 2140 pixels. The image processing system consisted of a Dell 7920 tower workstation (Dell Inc., Round Rock, TX, USA) with Windows10, a 3.0-GHz Intel® Xeon(R) Gold 6154, 256 GB RAM, and 2 NVIDIA GeForce RTX 2080Ti of 11 GB. Camera 1 and camera 2 were arranged on the inner wall of the fishpond, and the shooting angle was 90 degrees, covering the whole feeding area. Camera 3 was arranged on the bottom of the fishpond, and the water surface was shot from bottom to top. The three cameras all shoot 30FPS videos, and the image size of each frame is 3840 pixels × 2140 pixels. The 3 cameras and the feeding system are, respectively, connected to the computer, images captured are transmitted to the computer for processing, and then the computer sends instructions to control the feeding machine.

During the image acquisition, camera 1 and camera 2 were, respectively, attached to the inner wall of the fishpond at a position of 5–10 cm below the water surface, and the camera angles intersected vertically to form an acquisition area. Camera 3 is located at the bottom of the pond and looks up at the water's surface. Every 300 s, the feeding machine

above the pond dispenses feed pellets into the water surface evenly; the 3 cameras shoot the whole feeding process for 7 days.

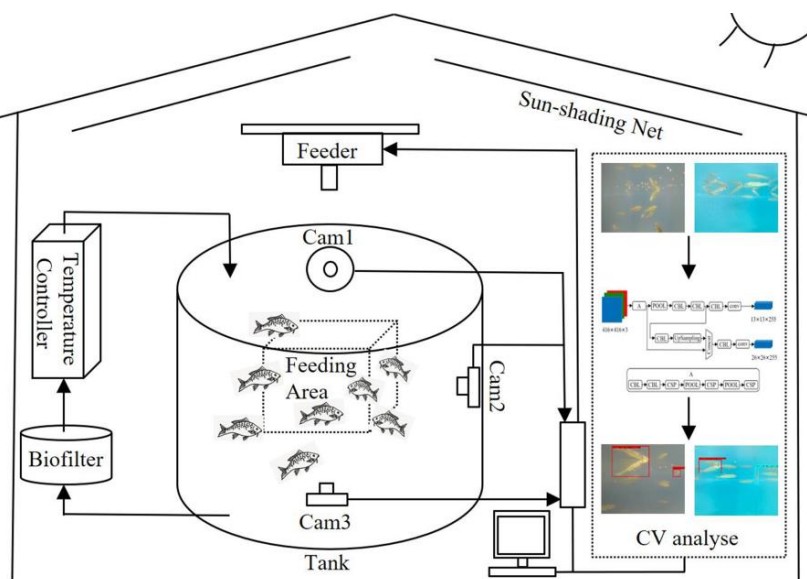

**Figure 1.** Apparatus for imaging of fry feeding.

### 2.3. Fry Feeding Status Judged by Manual Image Processing Methods

#### 2.3.1. Two Typical Characteristics of Fry Feeding

It can be known from the fry feeding images acquired during the adaptation period that the fries were young, and their food intake was small, so the disturbance amplitude of the water surface caused by feeding was little. When feeding, most fries swim up and touch the water lightly; while the cameras were set at a shallow underwater layer, the images captured would have a mirror effect because of the water's surface, as shown in Figure 2. In order to facilitate the study of fry feeding behavior, two typical characteristics of fry feeding are defined in this paper: angle feeding and vertical feeding, and their specific characteristics are described as follows:

(1) Angle feeding: fry swim up to the surface at an angle, and their mouths touch the feed floating on the surface, and also touch the fry in the water mirror, showing a shape of ">" or "<" in the image taken by the underwater camera, as shown in the figure below.

(2) Vertical feeding: fry swim up to the surface vertically, and their mouths touch the feed floating on the surface, and also touch the fry in the water mirror, showing a shape of "|" in the image taken by the underwater camera, as shown in the figure below.

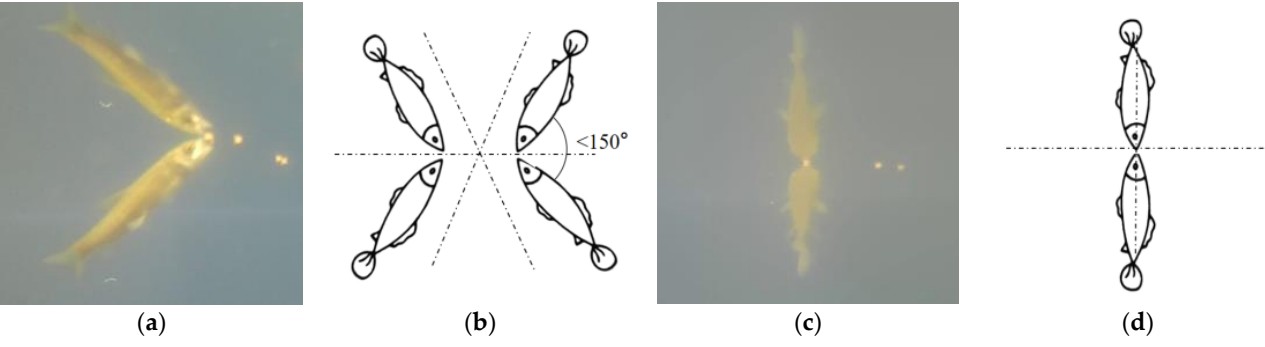

| (a) | (b) | (c) | (d) |

**Figure 2.** Two typical characteristics of fry feeding: (**a**,**b**) angle feeding images; (**c**,**d**) vertical feeding images.

### 2.3.2. Processing of Underwater Acquired Images

As shown in Figure 3, fry-feeding images were taken shallow underwater under different light conditions, (a) is the image acquired in sunny conditions and (b) is the image acquired in cloudy conditions. In this section, fry-feeding behaviors in the images were observed and counted manually.

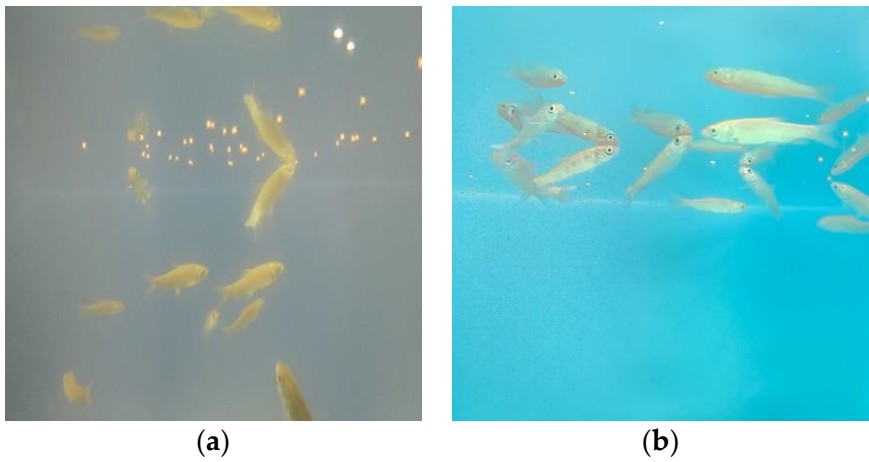

| (a) | (b) |

**Figure 3.** Images of fry feeding acquired in shallow underwater: (**a**) image acquired in sunny conditions; (**b**) image acquired in cloudy conditions.

As shown in Figure 4, images of fry feeding were taken at the bottom of the pond: (a) is the image of floating uneaten pellets and (b) is the image of fry feeding behavior. In this section, the feed pellets in the images were observed and counted manually.

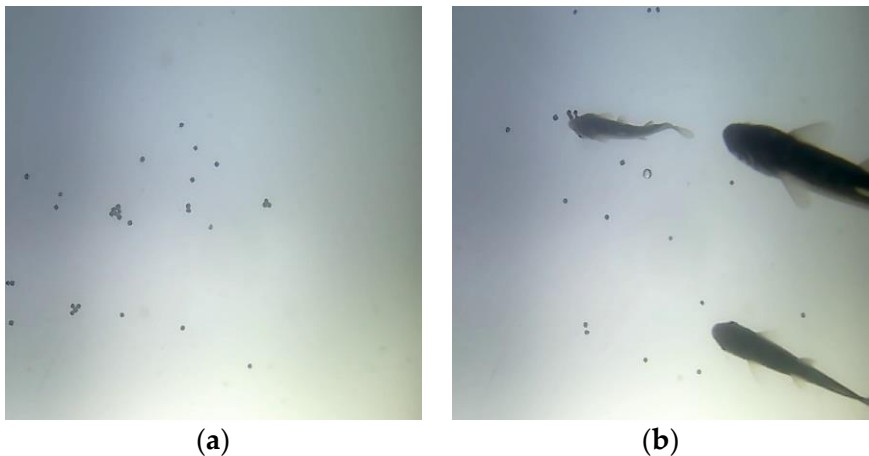

| (a) | (b) |

**Figure 4.** Images acquired from the bottom: (**a**) floating uneaten pellets; (**b**) fry feeding behavior.

### 2.3.3. Judging the Fry Feeding Status Manually

Figure 5 shows the relationship between fry feeding behaviors and uneaten pellets on the water surface based on the recognition of human observation. As can be seen from the figure, the number of uneaten pellets on the water surface decreases with the occurrence of fry feeding behaviors, and the higher the frequency of fry feeding characteristics detected, the faster the amount of uneaten pellets decreases; therefore, it proves that the occurrence of fry feeding behaviors can be detected based on fry feeding characteristics.

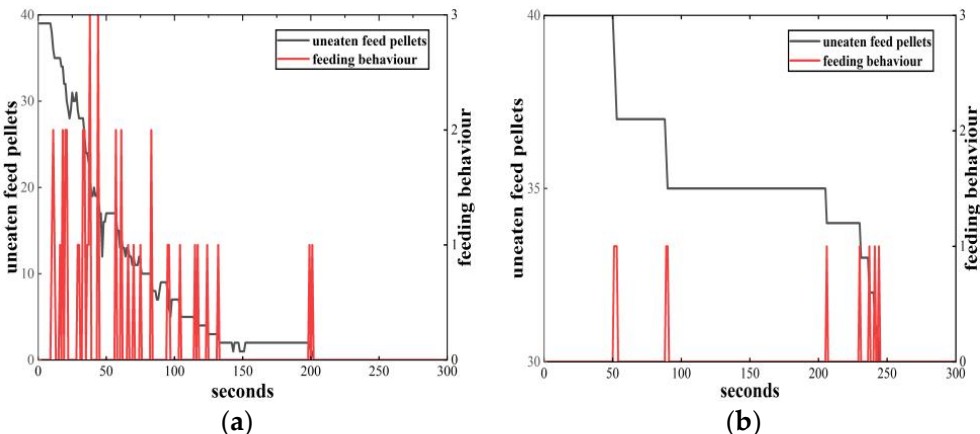

**Figure 5.** The relationship between fry feeding behaviors and uneaten pellets: (**a**) fry feeding in high intensity; (**b**) fry feeding in low intensity.

*2.4. Fry Feeding Status Judged by Computer Vision Image Processing Methods*

In this section, 2000 images acquired in the shallow underwater layer were selected for processing. The JPG format images were processed and made into data sets for the network training; the processing process is shown in Figure 6.

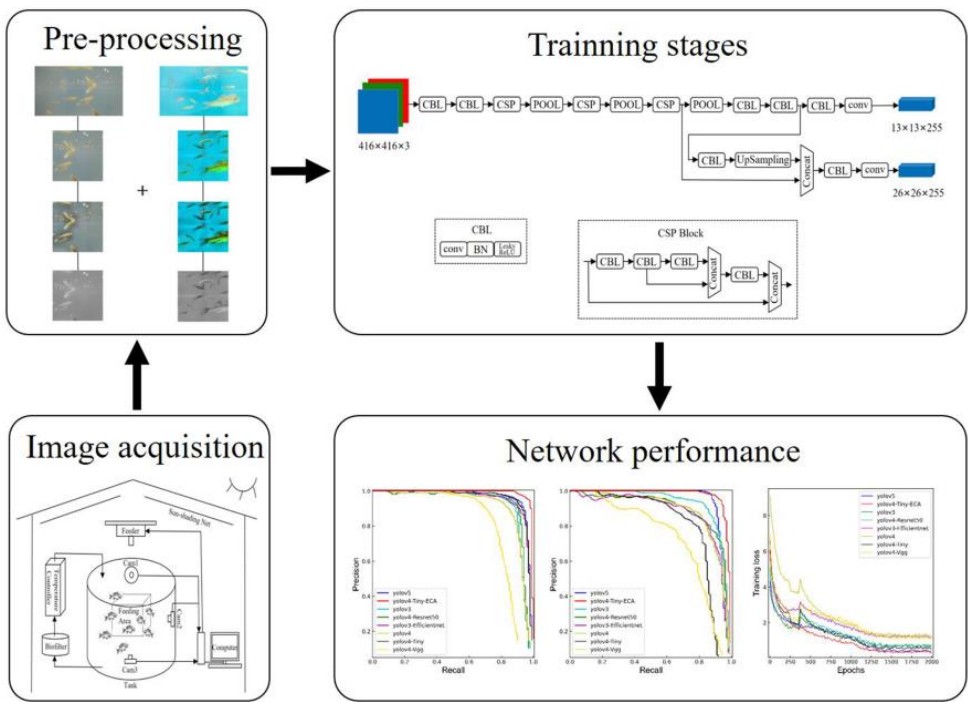

**Figure 6.** Schematic diagram of image acquisition, image preprocessing, and model training.

2.4.1. Image Preprocessing

In order to simulate the influence of different light conditions in the actual breeding and production site, the experimental system was arranged on the rooftop in this study; therefore, the external environment of the image acquisition includes soft light conditions and hard light conditions. In this study, the selected images were cut to obtain a square image containing the fry feeding characteristics, and then the images were processed with information enhancement, as shown in Figure 7.

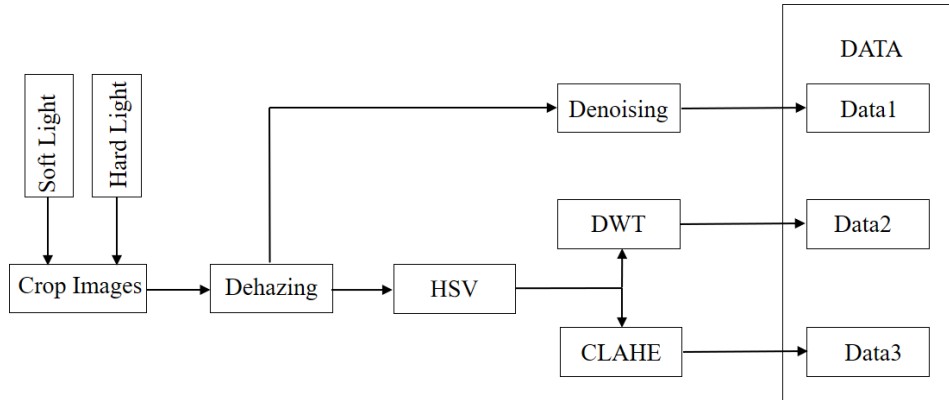

**Figure 7.** Process of image information enhancement and data set preparation.

　　　Images taken underwater are often affected by fog; therefore, we first defogged the images and then carried out three steps of processing the defogged images: (1) The images were denoised by median filtering (the window size is 64 × 64), which can remove the high-frequency noise in the image and make the image smoother. (2) The color channel conversion was carried out on the defogged images, images were converted from RGB format to HSV format, and then wavelet transform was carried out on the transformed HSV images, and finally converted them back to RGB images. The processed images can display texture information more clearly. (3) Color channel conversion was carried out on the defogged image, images were converted from RGB format to HSV format, and then the HSV images were separated by channel. The separated V-channel images were processed by CLAHE [29], and then the image channels were merged and converted back to RGB images, which can improve the clarity of the image. The data set obtained in the above three steps was combined to generate a network training data set containing 6000 images. The data set was labeled according to the fry feeding characteristics mentioned above (angle feeding and vertical feeding), and the labeling software was LabelImg (v1.8.1, https://pypi.org/project/labelImg/1.8.1/, accessed on 10 October 2022).

　　　Six thousand images are far from sufficient for training a robust model; in order to ensure accuracy, we used data augmentation to enlarge the dataset size and prevent overfitting. Methods include image rotation, flip, random changes in brightness, and random noise. The expanded data set and annotation files were sorted out, and files with poor quality were excluded. A dataset containing 58,360 images was obtained, among which 52,524 were training data sets and 5836 were verification data sets.

### 2.4.2. YOLOv4-Tiny Algorithm

　　　As a current excellent one-stage target detection algorithm, the YOLOv4 network represents continued improvements to previous YOLO series networks. It has a faster detection speed while guaranteeing network accuracy, and it can perform real-time detection on devices with low computing power. The YOLOv4-Tiny algorithm is designed on the basis of the YOLOv4 algorithm [30]. Its network structure is more simplified, and the number of network layers is reduced from 162 to 38, which greatly improves the target detection speed; however, its accuracy and speed can still meet the requirements of practical applications. The YOLOv4-Tiny algorithm uses the CSPDarknet53-Tiny network as the backbone network and uses the CSPBlock module in the cross-stage partial network [31] to replace the ResBlock module in the residual network. The CSPBlock module divides the feature map into two parts and merges the two parts through the CSP residual network so that the gradient information can be transmitted in two different paths to increase the correlation of gradient information.

　　　Compared with the ResBlock module, the CSPBlock module can enhance the learning ability of the convolutional network. Although the computation amount increases by 10–20%, its accuracy is greatly improved. In order to reduce the amount of computation,

the YOLOv4-Tiny algorithm eliminates the neck part with a high amount of computation in the original CSPBlock module, which not only reduces the amount of computation but also improves the accuracy of the algorithm. In order to further reduce the amount of computation and improve the real-time performance of the network, YOLOv4-Tiny uses the LeakyReLU function to replace the Mish function as the activation function [32]. The LeakyReLU function is shown in Formula (1):

$$y_i = \begin{cases} x_i, x_i \geq 0 \\ \frac{x_i}{a_i}, < 0 \end{cases} \tag{1}$$

Here $a_i \in (1, +\infty)$ is a constant.

In feature extraction and fusion, YOLOv4-Tiny is different from YOLOv4, which adopts spatial pyramid pooling (SPP) [33] and path aggregation network (PANet) [34] for feature extraction and fusion. The purpose of SPP is to increase the receptive field of the network, while PANet shortens the distance between features with large sizes at the bottom and features with small sizes at the top, making feature fusion more effective. The YOLOv4-Tiny algorithm uses a feature pyramid network (FPN) [35] to extract feature maps with different proportions to improve target detection speed. Meanwhile, it uses two different proportion feature maps (13 × 13 and 26 × 26) to predict detection results. The network structure of YOLOv4-Tiny is shown in Figure 8.

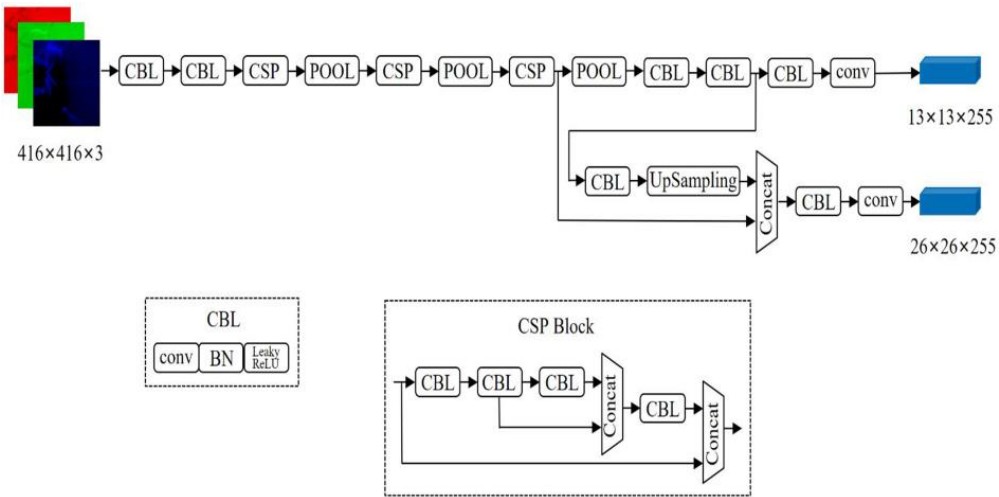

**Figure 8.** Structure of YOLOv4-Tiny model.

2.4.3. Calculation Method of Loss Function

The loss function $L$ of the YOLOv4-Tiny network consists of three parts, namely, bounding box regression loss $L_{Coord}$, confidence loss $L_{Conf}$, and classification loss $L_{Cls}$. The loss function $L$ can be expressed in Formula (2):

$$L = L_{Coord} + L_{Conf} + L_{Cls} \tag{2}$$

Bounding box regression loss $L_{Coord}$ can be expressed in Formula (3):

$$L_{Coord} = \lambda_{coord} \sum_{i=0}^{N \times N} \sum_{j=0}^{M} I_{ij}^{Obj} (2 - w_i \times h_i) [L_{CIoU}] \tag{3}$$

Confidence loss $L_{Conf}$ can be expressed in Formula (4):

$$L_{Conf} = -\sum_{i=0}^{N \times N} \sum_{j=0}^{M} I_{ij}^{Obj} [\hat{c}_i \log(c_i) + (1 - \hat{c}_i) \log(1 - \hat{c}_i)] - \\ \lambda_{noObj} \sum_{i=0}^{N \times N} \sum_{j=0}^{M} I_{ij}^{noObj} [\hat{c}_i \log(c_i) + (1 - \hat{c}_i) \log(1 - \hat{c}_i)] \tag{4}$$

Classification loss $L_{Cls}$ can be expressed in Formula (5):

$$L_{Cls} = -\sum_{i=0}^{N \times N} I_{ij}^{Obj} \sum_{c \in Classes} [\hat{p}_i(c) \log(p_i(c)) + (1 - \hat{p}_i(c)) \log(1 - p_i(c))] \tag{5}$$

The original image is divided into $N \times N$ grids, each of which generates $M$ candidate bounding boxes, forming a total of $N \times N \times M$ bounding boxes. If there is no target in the bounding box ($noObj$), then only confidence loss is calculated. Here, $i$ is the $i$-th grid of feature map, $j$ is prediction bounding box responsible for the $j$-th box, $w$ and $h$ are the width and height of the ground truth box. $I_{ij}^{Obj}$ represents whether there is a target in the $i$-th cell. $I_{ij}^{noObj}$ represents whether there is no target in the $i$-th cell. Bounding box regression loss will be multiplied by $(2 - w \times h)$ to enlarge the loss of the small bounding box. The confidence and classification loss function is cross-entropy loss. The confidence loss has two cases: any object and no object. In the case of no object, $\lambda$ is introduced to describe the loss percentage. $L_{CIoU}$ is denoted as the $CIoU$ loss between the prediction box and ground truth box. $L_{CIoU}$ can be expressed in Formulas (6)–(8):

$$L_{CIoU} = 1 - IoU + \frac{\rho^2(b, b^{gt})}{c^2} + \alpha v \tag{6}$$

$$v = \frac{4}{\pi^2} \left( arctan \frac{w^{gt}}{h^{gt}} - arctan \frac{w}{h} \right)^2 \tag{7}$$

$$\alpha = \frac{v}{(1 - IoU) + v} \tag{8}$$

where $IoU$ represents the ratio of intersection and union in two boxes. $\rho$ is the Euclidian distance of the two centers of the predicted box $(b)$ and ground truth box $(b^{gt})$, and $c$ is the diagonal length of the smallest box containing two boxes. $\alpha$ is the weight coefficient and the consistency of the aspect ratio is recorded as $v$. $w^{gt}$ and $h^{gt}$ are the width and height of the ground truth box and $w$ and $h$ are the width and height of the predicted box, respectively.

2.4.4. Attention Mechanism

In order to obtain a better training effect, the attention mechanism is introduced into the neural network in this study, and the performance of the original network model can be greatly improved by only adding a small amount of computation. The attention mechanism adopted in this paper includes CBAM module, SE module, and ECA module [36–38].

CBAM module is composed of a channel attention module and a spatial attention module in series. It calculates from the two dimensions of channel and space. Then, the attention map is multiplied by its input feature map for adaptive learning, so as to improve the accuracy of feature extraction by the model. The schematic diagram of its implementation is shown in Figure 9.

SE module is shown in Figure 10. After global average pooling (GAP) for each channel, the Sigmoid function is selected as its activation function through two nonlinear full connection layers. As can be seen from the figure, two nonlinear fully connected layers in SE module are used to capture nonlinear cross-channel interactions and reduce the complexity of the model through dimensionality reduction.

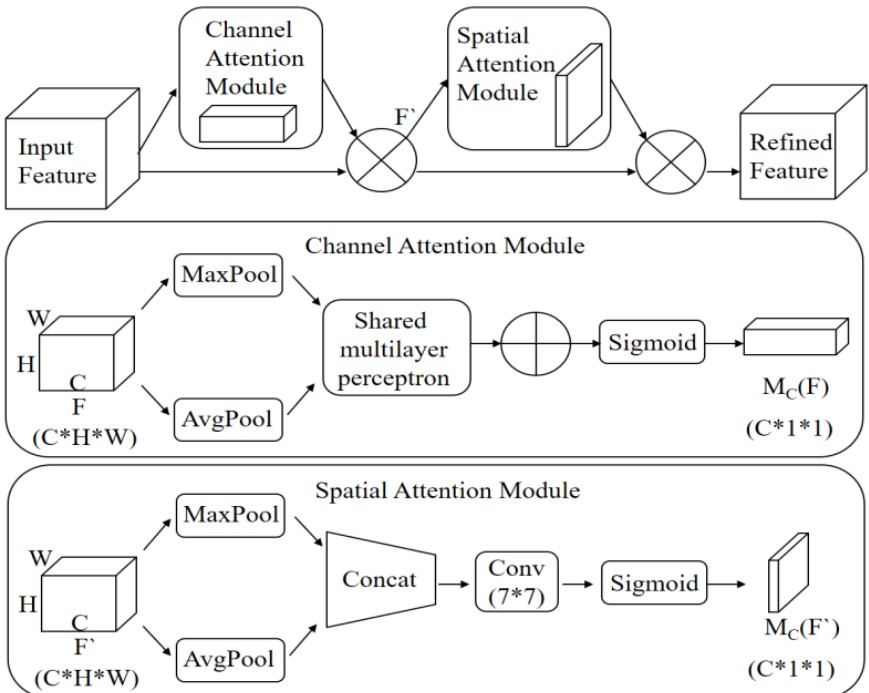

**Figure 9.** Structure of CBAM module.

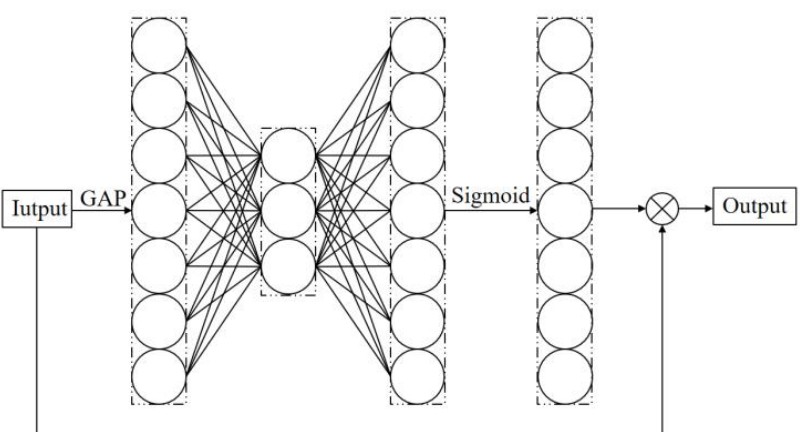

**Figure 10.** Structure of SE module.

The structure of the ECA module is shown in Figure 11. Different from the SE module, the ECA module also uses global average pooling but does not reduce channel dimension. ECA module adopts one-dimensional fast convolution with a convolution kernel size of k. K is not only the size of the convolution kernel but also represents the coverage of local cross-channel interaction, which means that k adjacent channels are involved in attention prediction near each channel. ECA module not only ensures the efficiency of the model but also ensures the effect of calculation. K value can be determined adaptively by the function of total channel number G, and the calculation is shown in Formula (9):

$$k = \varphi(G) = \left| \frac{\log_2 G + 1}{2} \right|_{odd} \tag{9}$$

where $k$ is the neighborhood of each channel; G is the total number of channels; $|x|_{odd}$ is the odd number closest to $x$.

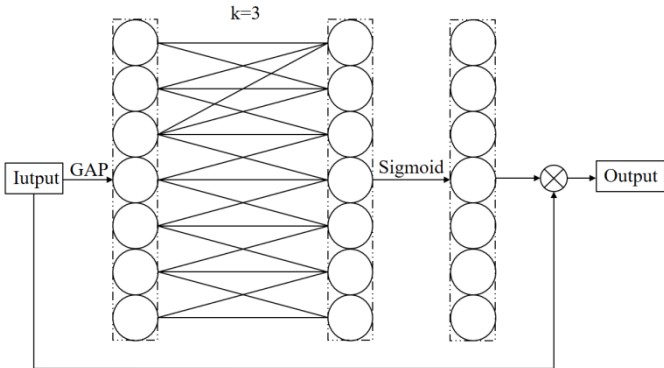

**Figure 11.** Structure of ECA module.

### 2.4.5. Model Performance Analysis

The training parameters used in this paper are shown in Table 1. The training stops when the verification loss remains constant for 10 epochs, and the learning rate increases by 0.1 times when the verification loss remains constant for 2 epochs.

**Table 1.** Network training parameters.

| Parameter | Value |
|---|---|
| Batch | 32 |
| Optimization method | Stochastic gradient descent |
| Initial learning rate | 0.001 |
| Momentum | 0.9 |
| Weight decay | 0.0005 |
| Training epochs | 2000 |

In this study, precision rate, recall rate, and $F_1$ score are used to measure the performance of the detection network. The precision rate refers to the proportion of correctly detected targets in the total number of detected targets, recall rate refers to the proportion of correctly detected targets in the total number of targets to be detected in the data set, and $F_1$ score can be regarded as a harmonic average of model accuracy and recall. The calculation methods of precision rate, recall rate, and $F_1$ score are shown in Formulas (10)–(12):

$$Precision = \frac{TP}{TP + FP} \tag{10}$$

$$Recall = \frac{TP}{TP + FN} \tag{11}$$

$$F_1 = \frac{2 \times P \times R}{P + R} \tag{12}$$

where TP is the abbreviation of "true positive", which refers to the correctly identified feeding fry; FN is the abbreviation of "false negative", which refers to the missed feeding fry; FP is the abbreviation of "false positive", which refers to the fish fry incorrectly detected as feeding status.

## 3. Results

As shown in Figure 12, with the YOLOv4-Tiny model, fry with angle feeding characteristics are detected in red boxes, and fry with vertical feeding characteristics are detected in blue boxes. The precision rate, recall rate, and $F_1$ score of angle feeding was 90.76%, 86.96%, and 0.89. The precision rate, recall rate, and $F_1$ score of vertical feeding was 85.46%, 74.09%, and 0.79. The detection speed of the model was 116FPS and the model size was 22.4 MB.

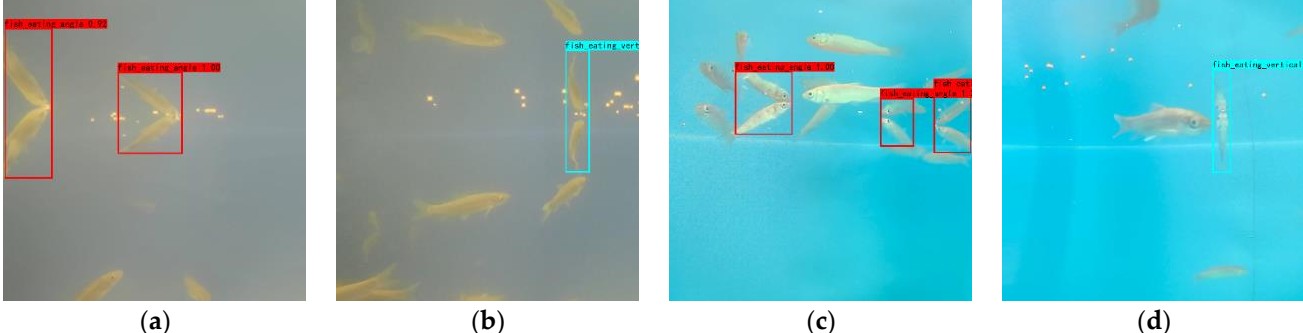

**Figure 12.** Model recognition effect. (**a**,**b**) model recognition effects on sunny days; (**c**,**d**) model recognition effects on cloudy days.

### 3.1. Comparison with Other Networks

This section compares YOLOv4-Tiny with six other commonly used advanced networks, such as YOLOv3, YOLOv3-EfficientNet, YOLOv4, YOLOv4-ResNet50, YOLOv4-VGG, and YOLOv5. All the networks in this paper were pre-trained on the Google ImageNet dataset, and all the comparison trials followed the same steps as YOLOv4-Tiny. To better illustrate the validity of this approach, a detailed comparison of the model results is presented in Table 2.

**Table 2.** Detection results of seven detection networks.

| Network | Precision (%) | | Recall (%) | | F1 score | | FPS | Model Size (MB) |
|---|---|---|---|---|---|---|---|---|
| | Angle | Vertical | Angle | Vertical | Angle | Vertical | | |
| YOLOv5 | 96.80 | 96.72 | 95.96 | 96.16 | 0.96 | 0.96 | 47 | 335 |
| YOLOv3 | 95.98 | 93.71 | 94.23 | 95.60 | 0.95 | 0.95 | 41 | 235 |
| YOLOv4-ResNet50 | 95.79 | 92.42 | 95.23 | 90.02 | 0.96 | 0.91 | 35 | 127 |
| YOLOv3-EfficientNet | 95.01 | 94.12 | 91.74 | 91.13 | 0.93 | 0.93 | 28 | 59.9 |
| YOLOv4 | 91.15 | 85.91 | 87.12 | 83.37 | 0.84 | 0.77 | 29 | 244 |
| YOLOv4-Tiny | 90.76 | 85.46 | 86.96 | 74.09 | 0.89 | 0.79 | 116 | 22.4 |
| YOLOv4-VGG | 86.79 | 85.43 | 67.96 | 63.20 | 0.76 | 0.73 | 43 | 89.9 |

Compared with the results of other models, the precision rate, recall rate, and $F_1$ score of the YOLOv4-Tiny model are in the middle and lower level, higher than YOLOv4-VGG and lower than other models; however, the processing speed of the YOLOv4-Tiny model is much higher than other models, and its model size is the smallest. By comparison, the YOLOv4-Tiny model has the potential to be applied to real-time detection.

### 3.2. Performance Comparison after Adding Attention Mechanism

As shown in Figure 13, three attention mechanism modules are added to the YOLOv4-Tiny network in this study, namely, CBAM module, SE module, and ECA module, respectively, and the adding positions are in the red star in the figure.

As shown in Table 3, the precision rate, recall rate, and $F_1$ score of the YOLOv4-Tiny network model with the addition of attention mechanisms are all improved. The detection performance of the YOLOv4-Tiny network model with the ECA module is not the highest, while its detection speed decreases at the least. Meanwhile, its detection performance is higher than those of other network models except YOLOv5 above. The precision rate, recall rate, and $F_1$ score of angle feeding was 96.23%, 95.92%, and 0.96. The precision rate, recall rate, and $F_1$ score of vertical feeding was 95.89%, 96.44%, and 0.96, respectively. At the same time, the detection speed decreased slightly to 108FPS; model size increased slightly to 22.7 MB.

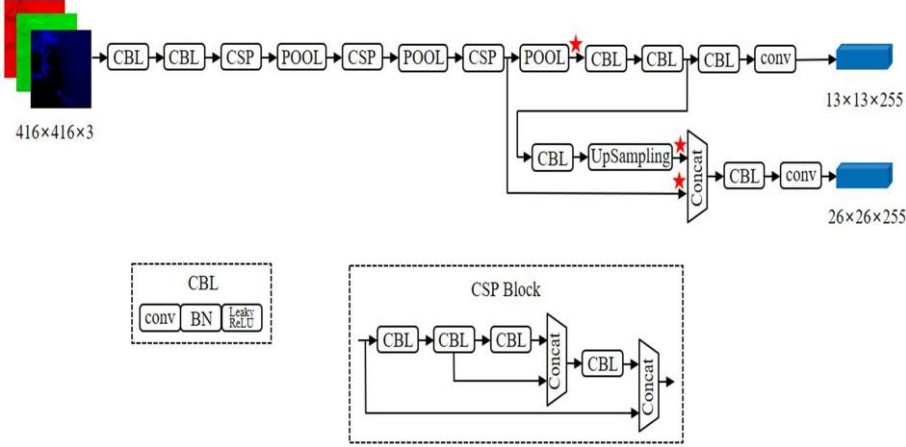

**Figure 13.** Structure of YOLOv4-Tiny model with attention mechanism added.

**Table 3.** Comparing network detection results.

| Attention | Precision (%) | | Recall (%) | | F1 Score | | FPS | Weight Size (MB) |
|---|---|---|---|---|---|---|---|---|
| | Angle | Vertical | Angle | Vertical | Angle | Vertical | | |
| None | 90.76 | 85.46 | 86.96 | 74.09 | 0.89 | 0.79 | 116 | 22.4 |
| CBAM | 96.43 | 95.34 | 95.84 | 96.44 | 0.96 | 0.96 | 91 | 22.4 |
| SE | 96.64 | 96.44 | 96.28 | 96.72 | 0.96 | 0.96 | 95 | 22.6 |
| ECA | 96.23 | 95.89 | 95.92 | 96.44 | 0.96 | 0.96 | 108 | 22.7 |

Figure 14 shows the precision-recall curves and loss curves of YOLOv3, YOLOv3-EfficientNet, YOLOv4, YOLOv4-ResNet50, YOLOv4-VGG, YOLOv5, and YOLOv4-Tiny-ECA network models. As can be seen from the figure, the precision–recall curve of the YOLOv4-Tiny-ECA model is closer to the upper right corner, and the region enclosed is the largest, indicating the best network detection effect. Meanwhile, it can be seen from the loss curves that the initial loss of the red curve of the YOLOv4-Tiny-ECA model decreases more slowly than other curves, which is due to the addition of an attention mechanism in the model. However, with the increase in training iterations, the red curve decreases faster and reaches a stable state earlier; therefore, the YOLOv4-Tiny-ECA model has better performance and a faster convergence rate.

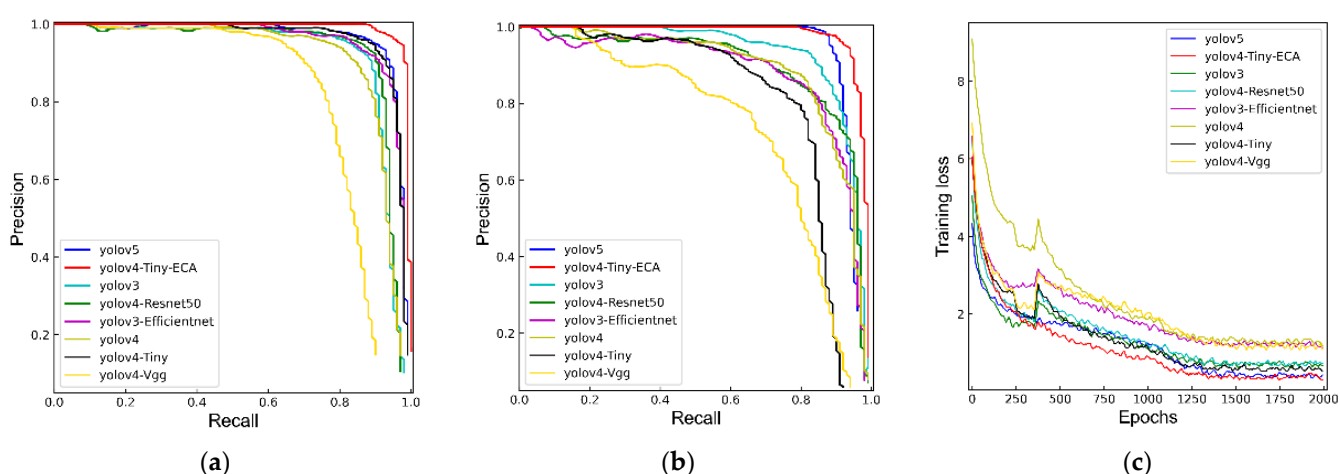

**Figure 14.** The precision–recall curves and loss curves: (**a**) precision–recall curves of angle feeding; (**b**) precision–recall curves of vertical feeding; (**c**) loss curves of different network models.

*3.3. Detection Performance Analysis under Different Underwater Shooting Conditions*

In order to verify the robustness of the YOLOv4-Tiny-ECA model, this paper tested the detection performance of the model under four different underwater shooting conditions. As shown in Figure 15, fry with angle feeding characteristics are detected in red boxes, and fry with vertical feeding characteristics are detected in blue boxes.

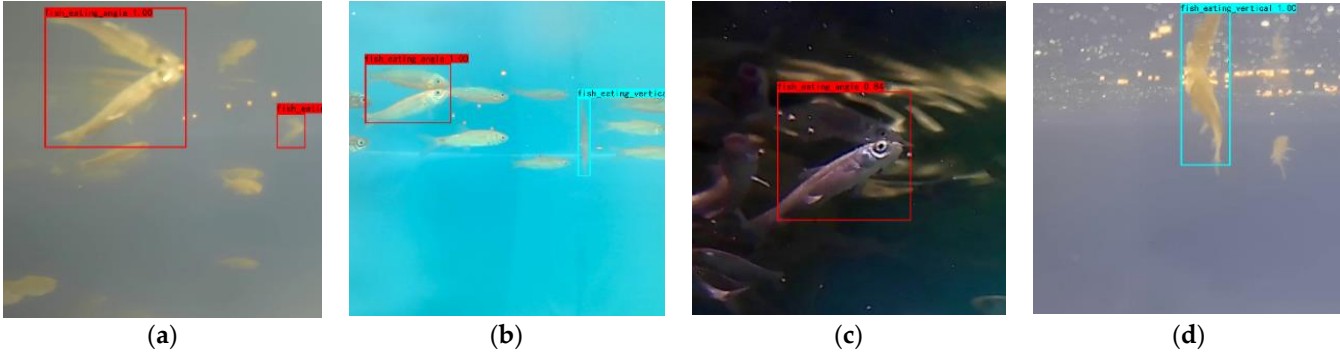

| (**a**) | (**b**) | (**c**) | (**d**) |

**Figure 15.** Network detection results under different conditions: (**a**) network detection results under sunny conditions; (**b**) network detection results under cloudy conditions; (**c**) network detection results under dark conditions; (**d**) network detection results under bubble conditions.

Table 4 shows the comparison of detection results under four different underwater shooting conditions. The results indicate that the YOLOv4-Tiny-ECA model can achieve good detection results under both sunny and cloudy conditions. The model detection performance was relatively weak under the condition of underwater bubbles, and the performance was not good in dark underwater conditions.

**Table 4.** Detection results under four different underwater shooting conditions.

| Environment | Precision (%) | | Recall (%) | | F1 Score | |
|---|---|---|---|---|---|---|
| | Angle | Vertical | Angle | Vertical | Angle | Vertical |
| Sunny | 94.62 | 95.33 | 95.84 | 94.72 | 0.97 | 0.95 |
| Cloudy | 96.40 | 96.85 | 96.64 | 96.78 | 0.98 | 0.96 |
| Dark | 75.79 | 74.07 | 69.44 | 62.86 | 0.68 | 0.60 |
| Bubble | 89.10 | 87.01 | 86.32 | 83.55 | 0.90 | 0.89 |

*3.4. Findings and Limitations*

In this study, we defined two typical characteristics (angle feeding and vertical feeding) of fry feeding and proved that the occurrence of fry feeding behaviors can be detected based on fry feeding characteristics. We also proposed a YOLOv4-Tiny-ECA network to detect the two typical characteristics from acquired images; then, we can realize variable feeding based on fry feeding status.

This study indicates that the YOLOv4-Tiny-ECA model has strong robustness to sunny daylight, cloudy daylight, and underwater bubble conditions and relatively weak detection results to dark underwater conditions, meaning that this method can be better applied to the detection of fry feeding status in the daytime.

## 4. Conclusions

In this study, we propose and verify a deep learning network for fry feeding status detection in factory farming, which will provide technical support for intelligent feeding. The results indicate that for the YOLOv4-Tiny-ECA model, the precision rate, recall rate, and $F_1$ score of angle feeding detection were 96.23%, 95.92%, and 0.96, respectively, and the precision rate, recall rate, and $F_1$ score of vertical feeding detection were 95.89%, 96.44%, and 0.96, respectively; the detection speed was 108FPS and the model size was 22.7 MB.

Considering the actual breeding environment, this study tested the detection performance of the model under the conditions of sunny, cloudy, dark, and bubble, and the results showed that the YOLOv4-Tiny-ECA model can achieve good detection results under sunny and cloudy conditions. The detection performance of the model is relatively weak under the condition of underwater bubbles and is not good under dark underwater conditions. It indicates that this method can be better applied to the detection of fry feeding status under natural light conditions in the daytime.

In future studies, our goal is to develop an effective feeding behavior detection system for multiple fry species in all-weather and all-season light conditions, which is more challenging but also more practical for application scenarios. We plan to obtain the feeding characteristics information of more species of fry, collect the feeding images of fry under different lighting conditions, and enhance the universality of the detection network. At the same time, we also plan to develop and deploy a set of edge computing detection equipment to realize real-time transmission of fry feeding status information. Due to the limitation of the computing capacity of the equipment, a more efficient and lighter-weight network is needed to realize the function. The ultimate purpose of fish feeding status detection is to serve intelligent feeding equipment, so we will also combine the feeding status detection function with variable feeding equipment to achieve 24-h precision fry feeding.

**Author Contributions:** Conceptualization, H.Y. and X.W.; methodology, H.Y. and X.W.; software, H.Y. and Y.S.; experiment, H.Y. and Y.S.; data collation, H.Y. and Y.S.; funding acquisition, Y.S. and X.W.; supervision, X.W.; writing—original draft, H.Y.; writing—review and editing, H.Y. and Y.S. All authors have read and agreed to the published version of the manuscript.

**Funding:** This work was supported financially by the Promotion Project of Modern Agricultural Machinery Equipment and Technology in Jiangsu Province (Grant No. NJ2020-09) and Jiangsu Science and Technology Planning Project of China (Grant No. BE2021362).

**Informed Consent Statement:** Not applicable.

**Data Availability Statement:** Not applicable.

**Acknowledgments:** The authors would like to thank Yao Wu for their technical and research support.

**Conflicts of Interest:** The authors declare no conflict of interest.

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
