# Peer review of "Detection Method of Fry Feeding Status Based on YOLO Lightweight Network by Shallow Underwater Images"

_electronics, doi:10.3390/electronics11233856_

Round 1

Reviewer 1 Report

1. The paper is well written and has a nice flow to it. The paper also provides a good overview of the existing methods and there is clarity about the contributions made.

2. Following clarifications/corrections are requested:

a) In line 123, change “Fish pond was sterilized before fry released.” to “Fish pond was sterilized before the fry population was released.”

b) How are the network training parameters given in Table 1 selected? Was there any optimization performed?

Author Response

Response to Reviewer 1 Comments

Dear Reviewers and Editor,

    We have revised the manuscript based on the Reviewers’ comments, and each change made in response to these comments is marked in red font in the revised manuscript (attached). Below, please find our responses to Reviewers’ comments:

 Comments and Suggestions for Authors

1. The paper is well written and has a nice flow to it. The paper also provides a good overview of the existing methods and there is clarity about the contributions made.

Response:

Thank you for your comments, we will continue to improve the network performance, try to realize the 24-hour feeding detection of fry in the future, and combine the feeding status detection function with variable feeding equipment to achieve precision feeding of fry.

2. Following clarifications/corrections are requested:

a) In line 123, change “Fish pond was sterilized before fry released.” to “Fish pond was sterilized before the fry population was released.”

b) How are the network training parameters given in Table 1 selected? Was there any optimization performed?

Response:

a) The manuscript has been revised and marked in red font. (line 123)

b) Parameters in Table 1 are selected by referring to relevant studies and pre-experiment. Batch size is selected according to the computer hardware capability. Training epochs is based on the results of the pre-experiment. We find that the curves tend to be flat after 2000epochs, with little decline. The other parameters are selected according to the appropriate parameters in the relevant detection network studies.

    The manuscript has been revised according to the expert's comments, and answered the relevant questions. The whole text has been checked and revised uniformly. The language of the revised manuscript has been re-edit by MDPI English Editing. Please review it again. Thank you very much.

Yours sincerely,

Haihui Yang,

College of Engineering, Nanjing Agricultural University, China

Reviewer 2 Report

The Abstract needs minor inclusion of YOLOv4 usage and prediction accuracy

The Highlights of the proposed system and the fry feeding status is not clear

The motivation towards selecting the YOLOV4 need to be mentioned clearly

Figure 1. Apparatus for imaging of fry feeding. In this figure what does that specifies computer pl interchange with the process it carries out , since the figure should be professional

What type of Fish fry is taken for experimentation and whether the different species are tried

Figure 5 is not clear pl enhance the low and high intensity process

Figure 6 redraw it

What is the purpose of Dehazing and Denoising

Equation 3 and 4 the parameters are confusing pl specify the usage of different parameters in a table and specify well

Pl avoid generic diagram like fig 8,9

In Table 1 the optimisation method to be in full form

As per the result does YOLOV4 VGG good results, have u tried with Yolov5 in different epochs

Author Response

Response to Reviewer 2 Comments

Dear Reviewers and Editor,

    We have revised the manuscript based on the Reviewers’ comments, and each change made in response to these comments is marked in red font in the revised manuscript (attached). Below, please find our responses to Reviewers’ comments:

 Comments and Suggestions for Authors

1. The Abstract needs minor inclusion of YOLOv4 usage and prediction accuracy

Response:

    The manuscript has been revised and marked in red font. (Abstract)

2. The Highlights of the proposed system and the fry feeding status is not clear

Response:

    In this study, we defined two typical characteristics (Angle feeding and Vertical feeding) of fry feeding, and the proposed system is used to detect them. As can be seen from figure 5, it proves that the occurrence of fry feeding behaviors can be detected based on fry feeding characteristics.

    The proposed system is based on computer vision image processing methods, the frequency of fry feeding behaviors can reflact fry feeding status. This manuscript is more focused on the detection performance of the network, the proposed method can provide technical support for intelligent feeding in factory fry breeding with natural light.

3. The motivation towards selecting the YOLOV4 need to be mentioned clearly

Response:

    As a current excellent one-stage target detection algorithm, the YOLOv4 network represents continued improvements to previous YOLO series networks. It has a faster detection speed while guaranteeing network accuracy, and it can perform real-time detection on devices with low computing power.

    The manuscript has been revised and marked in red font. (line 245)

4. Figure 1. Apparatus for imaging of fry feeding. In this figure what does that specifies computer pl interchange with the process it carries out , since the figure should be professional

Response:

    Figure 1 has been revised.

    We add the process of computer vision operation in the figure.

5. What type of Fish fry is taken for experimentation and whether the different species are tried

Response:

    In this study, 200 grass carp fry were taken for experimentation, with a mean weight of 5.5 g and mean body length of 7.4 cm. The fry were selected because of their strong adaptability and their wide use in aquaculture in China.

    In the future, the feeding characteristics of more fry species will be studied, such as tilapia and Largemouth BASS, the robustness of the detection network will be enhanced in combination with the all-weather and all-season light conditions, so as to realize the 24-hour feeding detection of fry.

6. Figure 5 is not clear pl enhance the low and high intensity process

Response:

    Figure 5 has been revised.

7. Figure 6 redraw it

Response:

    Figure 6 has been revised.

8. What is the purpose of Dehazing and Denoising

Response:

    In this study, cameras were arranged underwater, it is inevitable that there will be some floating objects and flocculent suspended objects in the fish pond. As can be seen from the images obtained from the pre-experiment, images taken underwater are blurry and affected by fog, so we defogged the images, and then carried out three steps of processing the defogged images.

    The preliminary experiment showed that, after the processing of Dehazing and Denoising, the feeding image obtained was clearer, the feeding characteristics of the fry were easier to observe, and the training precision was also guaranteed.

9. Equation 3 and 4 the parameters are confusing pl specify the usage of different parameters in a table and specify well

Response:

    The Equations to calculate loss function have been revised and marked in red font.

10. Pl avoid generic diagram like fig 8,9

Response:

    Figure 8 and figure 9 have been revised.

11. In Table 1 the optimisation method to be in full form

Response:

    SGD: Stochastic gradient descent

12. As per the result does YOLOV4 VGG good results, have u tried with Yolov5 in different epochs

Response:

    In this study, we selected 7 YOLO series networks (Table 2) and we found that the YOLOv4-Tiny network has the most potential to be implemented into actual production (small volume and fast calculation speed). Three different attention mechanisms were added to YOLOv4-Tiny network, and the detection results were compared, the YOLOV4-TINY-ECA network was selected.

    The YOLOv5 network was also tested and analyzed in this study, and it is found that although its detection accuracy is very high, its detection speed is not as good as that of YOLOv4-Tiny network, and it has the largest volume, so it is not suitable for deployment on edge devices at the current stage.

    In the future, we will try to adjust the YOLOv5 network, reduce its size as much as possible to verify whether it has the ability to detect the fry feeding status.

    The manuscript has been revised according to the expert's comments, and answered the relevant questions. The whole text has been checked and revised uniformly. The language of the revised manuscript has been re-edit by MDPI English Editing. Please review it again. Thank you very much.

Yours sincerely,

Haihui Yang,

College of Engineering, Nanjing Agricultural University, China

Reviewer 3 Report

The manuscript entitled Detection method of fry feeding status based on YOLO lightweight network by shallow underwater images investigated a very good practical research problem. The introduction of the background, motivation, and YOLO technology constraints is a good way. The related work presented the existing YOLO technology efforts and underwater images with objective detection techniques. The systematic diagram as shown in figure 6 is more optimal than existing studies. Simulation results shown the proposed work is more optimal than existing studies. However, still paper should be improved more.

1. Time complexity of the methods have not defined in the manuscript.

2. Data and workloads not cleared

3. There should be comparing table in related work.

4. Case studies related YOLO must be defined in manuscript

5. Finding and limitations must be added in the manuscript

6. The future direction discussed must be defined in the clear way

Author Response

Response to Reviewer 3 Comments

Dear Reviewers and Editor,

    We have revised the manuscript based on the Reviewers’ comments, and each change made in response to these comments is marked in red font in the revised manuscript (attached). Below, please find our responses to Reviewers’ comments:

Comments and Suggestions for Authors

The manuscript entitled Detection method of fry feeding status based on YOLO lightweight network by shallow underwater images investigated a very good practical research problem. The introduction of the background, motivation, and YOLO technology constraints is a good way. The related work presented the existing YOLO technology efforts and underwater images with objective detection techniques. The systematic diagram as shown in figure 6 is more optimal than existing studies. Simulation results shown the proposed work is more optimal than existing studies. However, still paper should be improved more.

1. Time complexity of the methods have not defined in the manuscript.

Response:

    During the image acquisition, the feeding machine above the pond put feed pellets into the water surface evenly for 300s, the 3 cameras shoot the whole feeding process for 7 days.

    In this study, we compared 7 YOLO series networks and found that the detection speed of the YOLOv4-Tiny model was 116FPS and the model size was 22.4MB, which indicated the potential of being applied to real time detection. (Table 2)

    Then we added 3 attention mechanism modules to YOLOv4-Tiny network, the detection performance of YOLOv4-Tiny-ECA network is better, with the detection speed decreased slightly to 108FPS, model size increased slightly to 22.7MB. (Table 3)

    This manuscript is focused on the detection speed and accuracy of the network, the operation time of the feeding equipment will be optimized in the future.

2. Data and workloads not cleared

Response:

    Figure 1 and figure 5 have been revised.

    In this study, the image acquisition system includes 3 waterproof cameras. During the image acquisition, camera 1 and camera 2 were respectively attached to the inner wall of the fish pond at a position of 5-10 cm below the water surface, and the camera angles intersected vertically to form an acquisition area. Camera 3 is located at the bottom of the pond and looks up on the water surface. Every 300s, the feeding machine above the pond put feed pellets into the water surface evenly, the 3 cameras shoot the whole feeding process for 7 days.

    We defined two typical characteristics (Angle feeding and Vertical feeding) of fry feeding, then we judged the fry feeding status manually, and found that the occurrence of fry feeding behaviors can be detected based on fry feeding characteristics. Then the proposed YOLOv4-Tiny-ECA network was used to detect the two typical characteristics from acquired images. Before network detection, images were Dehazed and Denoised.

3. There should be comparing table in related work.

Response:

    Table 3 has been revised.

    We compared the result of YOLOv4-Tiny with the results of YOLOv4-Tiny with the addition of attention mechanism modules.

4. Case studies related YOLO must be defined in manuscript

Response:

    YOLOv4-Tiny algorithm is defined in 2.5.2. Attention mechanism is defined in 2.5.4.

    In this study, figure 8 shows the structure of YOLOv4-Tiny model. Figure 13 shows the structure of YOLOv4-Tiny model with attention mechanism modules added, the adding positions are in the red star in the figure. Figure 9, figure 10 and figure 11 are the 3 attention mechanism modules.

    Figure 8, figure 9 and figure13 have been revised.

5. Finding and limitations must be added in the manuscript

Response:

    The manuscript has been revised and marked in red font, we add Findings and limitations in the manuscript. (line 432)

6. The future direction discussed must be defined in the clear way

Response:

    The manuscript has been revised and marked in red font, there are 3 future directions: (1) more species of fry; (2) lighter weight network; (3) variable feeding equipment.

    The manuscript has been revised according to the expert's comments, and answered the relevant questions. The whole text has been checked and revised uniformly. The language of the revised manuscript has been re-edit by MDPI English Editing. Please review it again. Thank you very much.

Yours sincerely,

Haihui Yang,

College of Engineering, Nanjing Agricultural University, China

Round 2

Reviewer 3 Report

The current version of the manuscript addressed all my previous comments.

I recommended this for further for accepting.